

# Sex- and tissue-specific expression of odorant-binding proteins and chemosensory proteins in adults of the scarab beetle *Hylamorpha elegans* (Burmeister) (Coleoptera: Scarabaeidae)

Angélica González-González[1,2], María E. Rubio-Meléndez[3], Gabriel I. Ballesteros[1,2], Claudio C. Ramírez[1,2] and Rubén Palma-Millanao[1,2]

[1] Centre in Molecular and Functional Ecology, Universidad de Talca, Talca, Chile
[2] Instituto de Ciencias Biológicas, Universidad de Talca, Talca, Chile
[3] Centro de Bioinformática y Simulación Molecular (CBSM), Facultad de Ingeniería, Universidad de Talca, Talca, Maule, Chile

Corresponding author
Rubén Palma-Millanao,
rupalma@utalca.cl,
rubenandrespalma@gmail.com

## ABSTRACT

In this study, we addressed the sex- and tissue-specific expression patterns of odorant-binding proteins (OBPs) and chemosensory proteins (CSPs) in *Hylamorpha elegans* (Burmeister), an important native scarab beetle pest species from Chile. Similar to other members of its family, this scarab beetle exhibit habits that make difficult to control the pest by conventional methods. Hence, alternative ways to manage the pest populations based on chemical communication and signaling (such as disrupting mating or host finding process) are highly desirable. However, developing pest-control methods based on chemical communication requires to understand the molecular basis for pheromone recognition/chemical perception in this species. Thus, with the aim of discovering olfaction-related genes, we obtained the first reference transcriptome assembly *of H. elegans*. We used different tissues of adult beetles from males and females: antennae and maxillary palps, which are well known for embedded sensory organs. Then, the expression of predicted odorant-binding proteins (OBPs) and chemosensory proteins (CSPs) was analyzed by qRT-PCR. In total, 165 transcripts related to chemoperception were predicted. Of these, 16 OBPs, including one pheromone-binding protein (PBP), and four CSPs were successfully amplified by qRT-PCR. All of these genes were differentially expressed in the sensory tissues with respect to the tibial tissue that was used as a control. The single predicted PBP found was highly expressed in the antennal tissues, particularly in males, while several OBPs and one CSP showed male-biased expression patterns, suggesting that these proteins may participate in sexual recognition process. In addition, a single CSP was expressed at higher levels in female palps than in any other studied condition, suggesting that this CSP would participate in oviposition process. Finally, all four CSPs exhibited palp-biased expression while mixed results were obtained for the expression of the OBPs, which were more abundant in the palps than in the antennae. These results suggest that these chemoperception proteins would be interesting novel targets for control of *H. elegans,* thus providing a theoretical basis for further studies involving new pest control methods.

## INTRODUCTION

Chemical perception in insects relies on olfaction and gustation, which enable insects to accomplish important tasks such as mating, host finding and predator avoidance in their environment (*Sánchez-Gracia, Vieira & Rozas, 2009*). Physically, these abilities depend on small, specialized organs called sensilla (*Mutis et al., 2014*), which populate insect's appendices, such as antennae and palps. Sensilla are exposed to chemical stimuli and protect the neuronal dendrites inside (*Suh, Bohbot & Zwiebel, 2014*). Between the cuticle and the dendrites is the sensillar hemolymph, which contains plenty of soluble proteins that carry chemical odorants (*Sánchez-Gracia, Vieira & Rozas, 2009*). The most important insect carrier proteins are chemosensory proteins (CSPs) and odorant-binding proteins (OBPs), both of which are responsible for recognizing (*Leal, 2013*; *Li et al., 2013*) and transporting volatiles, typically hydrophobic molecules (*Gadenne, Barrozo & Anton, 2016*), from the sensillar pores to the membrane receptors located in the dendrites of the neuron (*Pelosi et al., 2018*). These proteins are the first step in the cascade of events that compromise the olfactory and gustatory processes (*Leal, 2013*).

OBPs are small globular proteins (10–14 kDa), typically with acidic isoelectric points and are as abundant as 10 mM in the sensillar hemolymph. They are highly divergent in sequence (*Zhu, Zhao & Yang, 2012*) and primarily show six highly-conserved cysteine residues, called Classic OBPs, that combine to form three disulfide bonds conferring important functional characteristics (*Brito, Moreira & Melo, 2016*). Among the OBPs, there are other forms that have less than six cysteines (typically four or five) called Minus-C OBPs (*Spinelli et al., 2012*) while those with more than six cysteines are called Plus-C OBPs and those with exactly twelve cysteines are called Dimer OBPs (*Zhou et al., 2004*). In addition to this classification, OBPs can be distinguished from pheromone-binding proteins (PBPs) based on the type of volatile that they bind; PBPs recognize molecules emitted by conspecific individuals (*Mao et al., 2016*), with sexual pheromones being the most deeply studied. Finally, general odorant-binding proteins (GOBPs) bind general odorants (*Glaser et al., 2013*) such as host volatiles. In addition, CSPs are smaller than OBPs (8–11 kDa), and they show only four conserved cysteine residues. However, their sequences are more conserved than those of OBPs (*Zhu, Zhao & Yang, 2012*), and phylogenetically, CSPs seem to form a more homogeneous group of proteins (*Pelosi et al., 2018*).

Another important function of these carrier proteins is to protect odorant molecules from the odorant degrading enzymes (ODEs) found in the sensillar lymph (*Zhang, Walker & Wang, 2015*) before the odorant molecules reach the olfactory or gustatory receptors (ORs and GRs, respectively). These receptors belong to a large and diverse superfamily of seven-transmembrane-domain receptors (*Dahanukar, Hallem & Carlson, 2005*; *Iatrou & Biessmann, 2008*) expressed in the dendrites inside the sensilla (*Leal, 2014*) and trigger signal transmission through the neuron to the primary olfactory center of the brain (*Gadenne,*

*Barrozo & Anton, 2016*). Interestingly, ORs form a heterodimeric complex with an odorant receptor coreceptor (ORCo) (*Leal, 2013*). In contrast to ORs, ORCOs are highly conserved among insect species (*Dahanukar, Hallem & Carlson, 2005*); only one representative has been reported for each insect species (*Missbach et al., 2014*), and its presence is considered crucial for odorant perception (*Stengl & Funk, 2013*; *Zhou et al., 2014a*; *Zhou et al., 2014b*). Thus, these protein families are notoriously important for insects. However, until now, few studies have addressed the sex- and tissue-specific expression patterns of these proteins, especially in numerous groups of pest species among the coleopterans and particularly in the scarab beetles.

*Hylamorpha elegans* (Burmeister) is a widespread beetle species native to Chile and is also present in Andean proximities of Argentina's Neuquén province (*Ratcliffe & Ocampo, 2002*). As in other interesting cases reported from this family worldwide, this species has been able to adapt to the changes in its environment resulting from the replacement of native vegetation with exotic cultivated species (*Lefort et al., 2014*; *Lefort et al., 2015*). Thus, it has become an important pest in pastures, cereal and berry crops and common hazel orchards (*Durán, 1951*; *Cisternas, 1992*; *Cisternas, 2002*; *Cisternas, 2013*; *Aguilera et al., 1996*; *Aguilera, Guerrero & Rebolledo, 2011*) and is able to cause severe economic damage (*Artigas, 1994*). Despite the important environmental changes resulting from the introduction of exotic host plant species, *H. elegans* still has a close relationship with tree species from the genus *Nothofagus*. Most adult beetles use the canopy of these trees to congregate, feed and mate, and in some seasons, they cause severe defoliation in *Nothofagus* forests (*Bauerle, Rutherford & Lanfranco, 1997*; *Lanfranco et al., 2001*). As occurs in other scarab beetle species, due to cryptic position of the larvae in the soil and nocturnal activity of the adults, pest control is difficult using conventional methods, such as the use of insecticides applied to the soil (*Jackson & Klein, 2006*). Thus, alternative ways to manage the pest populations are highly desirable. Therefore, controlling the beetles based on ethology such as disrupting mating or host finding appears to be an interesting option in controlling this pest (*Quiroz et al., 2007*; *Venthur et al., 2016*). Although the chemical communication of the species has been studied, and the ability of males to perceive receptive females was reported (*Quiroz et al., 2007*), very little is known about the molecular basis of *H. elegans* olfaction and chemical communication. However, a recent report revealed an olfactory protein linked to the ability to perceive volatiles commonly used in chemical communication by Scarabaeidae species (*Venthur et al., 2014*; *González-González et al., 2016*), which suggests that a similar mechanism could be involved in chemical communication in *H. elegans*.

Considering this information, our goals were to identify the repertoire of proteins related to taste and olfaction in adults of *Hylamorpha elegans* by performing a transcriptome analysis and comparing the relative expressions of OBPs and CSPs in different tissues of both sexes; we hypothesized that OBPs and CSPs are highly expressed in the antennae and palps compared to their expression in a nonsensory tissue, and there are differential expression patterns of these transcripts between males and females. To our knowledge, this is the first report on the identification and characterization of multiple olfactory genes in the scarab beetle *H. elegans*.

## MATERIALS & METHODS

### Tissue collection

Males and females of *H. elegans* were collected from the field at the peak of their flight season (*Quiroz et al., 2007*). Males and females ($n = 120$ of each sex) were separated in fresh plastic tubes and anesthetized using cold ice packs. For each individual, both antennae and maxillary palps in addition to the right hindleg tibia were excised using clean tweezers and immersed in RNAlater Solution (Qiagen, Hilden, Germany) following the manufacturer's instructions to preserve RNA. Then, the samples were stored at $-80\ ^{\circ}C$ until RNA extraction.

### RNA extraction and sequencing

Total RNA was extracted grinding the tissue using sterile, disposable plastic pestles in liquid nitrogen, and using the RNeasy Plant Mini kit (Qiagen, Hilden, Germany) following the manufacturer's instructions for use in RNA sequencing and relative expression RT-qPCR assays. A total of 120 males and 120 females were used for RNA extractions, leading to six samples composed of 240 antennae, 240 maxillary palps and 120 right hindleg tibias from each sex. Total RNA was precipitated using 0.1 volumes of 3M sodium acetate and 2 volumes of 100% ethanol and shipped to Macrogen (Seoul, South Korea) for sequencing the six libraries. Polyadenylated mRNA was used for library construction using the TruSeq Stranded Total RNA Sample Preparation kit (Illumina, San Diego, CA, USA) and tagged, pooled and sequenced using an Illumina HiSeq 2000 ($2\times100$ bp, paired-end libraries). Two FASTQ files were created for each sex and tissue. Raw transcriptome data were deposited in NCBI's Sequence Read Archive database under ID: SRP137879.

### Assembly

Illumina RNA-seq libraries were quality checked with FastQC v0.11.3 (http://www.bioinformatics.babraham.ac.uk/projects/fastqc) to assess the presence of adapters derived from sequencing, overrepresented k-mers, read length and overall read quality scores. All libraries were processed with Trimmomatic v0.35 (*Bolger, Lohse & Usadel, 2014*) to remove any remaining TruSeq adapter sequences and to eliminate low quality bases (Q < 30) from reads. After sequence processing, all remaining sequences shorter than 36 bp long were also removed from all datasets. Clean Illumina datasets were pooled *in silico* by concatenating library files. Before assembly, ribosomal RNA reads were removed by mapping the libraries using Bowtie v1.1.1 (*Langmead et al., 2009*) against a custom rRNA database created from insect ribosomal sequences downloaded from NCBI and keeping unmapped reads. The remaining high-quality reads were *de novo* assembled with Trinity v2.0.6 (*Haas et al., 2013*) using default parameters. Metrics for *de novo* assembly were obtained with QUAST v2.3 (number of contigs, total length, N50, largest contig and %GC) (*Gurevich et al., 2013*), while transcriptome completeness was assessed by benchmarking the assembled transcriptome using BUSCO (Benchmarking Universal Single-Copy Orthologs) v3.0.2b (*Simão et al., 2015*) using the Insecta, Arthropoda and Endopterygota reference database. To determine whether this transcriptome encoded one or more sets of core genes conserved across a range of insect species, a "completeness score" was calculated (*Moreton, Izquierdo*

*& Emes, 2016*). A total of 1,658 near-universal single-copy orthologs from insect species were used as reference core genes (available at http://busco.ezlab.org; *Simão et al., 2015*).

## Annotation and functional gene classification

Homology searches of contigs from the assembled *de novo* transcriptome were performed locally with BLASTx using the NR database (NCBI) as a reference, setting an e-value of 1e−5 as a threshold. Automatic annotation using gene ontology (GO) terms was performed by loading the nucleotide FASTA file (Trinity transcriptome assembly) together with BLASTx results in XML format into Blast2GO v2.8 (*Conesa & Götz, 2008*). We also performed InterPro annotation, GO term assignment, and enzyme code and pathway annotation using the Kyoto Encyclopedia of Genes and Genomes (KEGG) terms integrated into Blast2GO. Successfully annotated transcripts were categorized and assigned to GO terms from different GO categories (molecular function, cellular component and biological process). The final contig annotation table (Fig. S1) was exported from Blast2GO.

## Putative chemoreception-related protein annotation

Identification of putative *H. elegans* chemosensory gene families was performed by aligning the assembled contigs with a local database of protein sequences of OBPs, CSPs, GRs, ORs, IRs, ORCo, SNMPs, ODEs and PDEs from 954 insect species, using BLASTx (*Altschul et al., 1997*). These references, known sequences were downloaded from NCBI (ncbi.nlm.nih.gov) and manually curated to build a non-redundant, reference database. Then, putative annotated transcripts were manually curated in order to eliminate isoforms, duplications and incomplete transcripts. Finally, complete-length sequences of the OBPs and CSPs in *H. elegans* were identified using the ORF finder tool (https://www.ncbi.nlm.nih.gov/orffinder/) (*Bin et al., 2017*).

## Phylogeny of OBPs and CSPs

Annotated complete length sequences for predicted OBPs and CSPs based on the amino acid sequences from *H. elegans*, *Anomala corpulenta*, *Holotrichia oblita*, *Dendroctonus ponderosae* and *Tribolium castaneum* were aligned using ClustalW (*Thompson, Higgins & Gibson, 1994*). Then, maximum-likelihood trees were built using CLC Workbench v7.8.1 (Qiagen, Hilden, Germany) using 1,000 replications.

## qRT-PCR relative expression analysis

Total RNA from three groups of males and three groups of females (ca. 40 individuals each) was treated to eliminate genomic DNA contamination using a Turbo DNA-free kit (Thermo Fisher, Vilnius, Lithuania) following the manufacturer's protocol. Then, cDNA was synthetized using an AffinityScript cDNA kit (Agilent Technologies, Santa Clara, CA, USA) using oligo(dT) primers according to the manufacturer's protocols. The expression of the OBP and CSP genes in *H. elegans* was detected by real-time PCR (qRT-PCR) using a Stratagene MX3000 thermocycler (Agilent Technologies, Santa Clara, CA). The cDNA amplification reactions were carried out with 5 ng of total cDNA and a SYBR Fast Universal qPCR kit (Kapa Biosystems, Wilmington, MA, USA). The reaction was performed according to the manufacturer's recommendations. Two technical replicates

for each of the three biological replicates were performed, and as a negative control, we used water in the reaction mixture instead of cDNA. The temperature profile used was 95 °C for 10 min; followed by 40 cycles of 95 °C for 15 s, 60 °C for 15 s, and 72 °C for 20 s. After each amplification step and at the end of the amplification, the fluorescence was measured. For gene expression analysis, specific primers were designed for the OBPs, CSPs (Table 1) and four reference genes (Table 2) from the *H. elegans* transcriptome. All primers were designed with Beacon Designer v8.13 software (Premier Biosoft, Palo Alto, CA, USA). The expression of each gene was normalized to the expression of the *EF1a* gene because a constant expression level was observed for *EF1a* in the different tissues of both sexes, and estimations of relative expression levels were made for each gene using the method of *Livak & Schmittgen (2001)*. Finally, statistical analysis was performed using analysis of variance (ANOVA) and the multiple range Tukey's test ($p \leq 0.05$). All analyses were performed with InfoStat v217.1.2 statistical software (*Di Rienzo et al., 2011*).

## RESULTS

### *H. elegans* reference transcriptome assembly

Using Trinity, filtered reads were assembled into 138,270 contigs (N50 of 2,189 bp, mean length of 789.3 bp). Among the assembled contigs, 49,752 (35.98%) were less than 300 bp in length, 60,126 (43.48%) contigs were between 300 bp and 1,000 bp in length, and 28,392 (20.53%) contigs had a size greater than 1,000 bp (Table 3). Completeness analysis using BUSCO with the Insecta reference database showed that 97.2% complete conserved genes were found in the assembly and 1.5% corresponded to fragmented conserved genes, while only 1.3% of the single-copy ortholog genes were missing (Table 4). An additional BUSCO analysis against Arthropoda and Endopterygota is shown in Table 4, with similar completeness values.

### Sequence annotation

The BLASTx alignments revealed that 51,950 contigs (37.57% of the total contigs) were annotated with known proteins within the NR database. Most transcript sequences with protein hits matched those of other beetles, such as *Oryctes borbonicus* and *Tribolium castaneum* (Fig. 1). Among the BLASTx-aligned contigs, 42,329 (30.61%) could be annotated based on their sequence homology with GO terms (Fig. 2). Additionally, among the total number of sequences analyzed ($N = 138,270$) with InterProScan (IPS), 22,889 (16.55%) sequences were assigned to at least one InterProScan/GO term. The remaining sequences were classified as ''IPS match but no GO terms'' (26,881; 19.44%) or as no IPS match (88,500; 64%) (Fig. S1).

### Chemosensitive protein annotation

The annotation process predicted 23 odorant-binding proteins, one pheromone-binding protein, four chemosensory proteins, 26 gustatory receptors, 70 odorant receptors, one odorant receptor coreceptor, three sensory neuron membrane proteins, 15 odorant-degrading enzymes, and 22 ionotropic receptors.

**Table 1  Identification, sequence length, primers sequences, and amplicon size of OBPs, CSPs, and PBP genes to *Hylamorpha elegans*.**

| Sequence ID | Sequence length (pb) | Forward primer sequences (5′–3′) | Reverse primer sequences (5′–3′) | Product length (pb) |
|---|---|---|---|---|
| **(a) *OBPs*** | | | | |
| *HePBP1* | 426 | TGATGATGGTATTATAGATG | GGTAGACATTATCACAAG | 128 |
| *HeleOBP1* | 432 | AGACAGACAGGAACGGATA | TATCTATAAGGGTCGGGTCAA | 77 |
| *HeleOBP2* | 477 | AAGTGCTGGTGCTACTAA | GGTCCTTCATCGCTTCTA | 167 |
| *HeleOBP3* | 552 | GCTGGATGTGTTTAAGTTTA | CGAATCTGCGTTGTAATG | 92 |
| *HeleOBP4* | 450 | GTGGCATACTGAATAGCA | TAGGTCTGTTGACAAGGA | 124 |
| *HeleOBP5* | 501 | GCAAGACAATAACGGTAA | CACGGTATCTAAGCAGTA | 121 |
| *HeleOBP6* | 449 | TACTTGCTTCACTTCTCC | GGTATTCATCGCTTGTTG | 80 |
| *HeleOBP7* | 507 | GCTTAGAATCATCCACAA | ATGCCTATACTTCGTAGA | 157 |
| *HeleOBP8* | 453 | GTTACCAGGATACAAGAAG | CGACTAGATTCCGTAGAT | 103 |
| *HeleOBP9* | 453 | TGGAAGCAGATAGCGATTA | GTCATCATCATCAGGAACAG | 80 |
| *HeleOBP11* | 471 | CCTTGCCTTGTGCTCATA | CCTGTGTCCATAATCTTCTCTAA | 84 |
| *HeleOBP13* | 414 | AATCGCTACCAGGAACAG | GTGACGCATACCGACATT | 100 |
| *HeleOBP14* | 693 | AAGAAGGAGATGAGATGTG | AGCCTAAATCAGCAGTAAT | 124 |
| *HeleOBP15* | 633 | GCTCAATCGTTAGAATGT | TGTGCTTCATCTTCATAAG | 161 |
| *HeleOBP17* | 447 | ATCAAATAACGAACCCTCTC | CCGCTGCTATTCAGTATG | 81 |
| *HeleOBP18* | 468 | ATAGTGGACGCCGTTAATG | CTGCCTGTTAGACCTTGAC | 77 |
| *HeleOBP19* | 384 | TCTCATAAGTGTGCGAAT | CGTAACTCCTTCCGTATA | 83 |
| *HeleOBP21* | 927 | TACGAGATGAAGGCGAAT | AATTGGCTGTAGGTGTAAG | 88 |
| **(b) *CSPs*** | | | | |
| *HeleCSP1* | 405 | TTGTTGTGTTAAGTGTTGT | CCAGAAGACGATCATTATTC | 114 |
| *HeleCSP2* | 390 | GGTGTTAGTTGTGTTAAGTGTT | CCTTCCAGAAGACGATCATTA | 123 |
| *HeleCSP3* | 390 | GTGTTAGTTGTGTTGAGT | CTTATTGTATCCTTCCAGAA | 132 |
| *HeleCSP4* | 264 | AGTATTGTCGTGGTAGTG | GCATGGTCCTTCATCTAA | 141 |

**Table 2  Primers sequences, and amplicon size for normalizer genes tested in *Hylamorpha elegans*.**

| Normalized Gene ID | Forward primer sequences (5′–3′) | Reverse primer sequences (5′–3′) | Product length (pb) |
|---|---|---|---|
| *He18S* | CGATGTCAGTGTGGATAC | TCAATGTTGTTCTGAATGC | 75 |
| *HeEF1a* | TCAAGCAACTTATTGTAGGT | TTAATGTACGACGAGACTTC | 103 |
| *HeB-Actin* | CTCTTCCAACCTTCATTCTT | GTCAACATCGCACTTCAT | 84 |
| *HeGADPH* | AATATCTACCAGGACATAA | ATGAGTATATCTGCTTCT | 85 |

## Quantitative real-time RT-PCR analysis

We investigated the expression patterns of 20 putative *H. elegans* OBP and CSP genes by performing a qRT-PCR analysis of the RNA extracted from adult male and female antennae, male and female tibias as well as male and female maxillary palp tissues (Fig. 3). The predicted *HelePBP1* in male antennae showed remarkably larger expression than that of the palps and tibias of both sexes but was not distinguishable from that of the female antennae (Fig. 3A). Among the remaining 15 tested OBPs, *HeleOBP2*, *HeleOBP5*, *HeleOBP8*, *HeleOBP9* and *HeleOBP21* also displayed antenna-specific expression (Figs. 3C,

**Table 3  _H. elegans_ transcriptome assembly statistics.**

| | |
|---|---|
| # contigs ($\geq$ 0 bp) | 138,270 |
| # contigs ($\geq$ 1,000 bp) | 28,392 |
| Total length ($\geq$ 0 bp) | 109,137,411 |
| Total length ($\geq$ 1,000 bp) | 66,930,846 |
| # contigs | 51,549 |
| Largest contig | 27,382 |
| Total length | 82,968,309 |
| GC (%) | 36.85 |
| N50 | 2,189 |
| N75 | 1,196 |
| # N's per 100 kbp | 0 |
| _H. elegans_ transcriptome assembly statistics | |
| # contigs ($\geq$ 0 bp) | 138,270 |
| # contigs ($\geq$ 1000 bp) | 28,392 |
| Total length ($\geq$ 0 bp) | 109,137,411 |
| Total length ($\geq$ 1,000 bp) | 66,930,846 |
| # contigs | 51,549 |
| Largest contig | 27,382 |
| Total length | 82,968,309 |
| GC (%) | 36.85 |
| N50 | 2,189 |
| N75 | 1,196 |
| # N's per 100 kbp | 0 |

**Table 4  BUSCO analysis on _H. elegans_ assembly.**

| BUSCO category | Insecta | Endopterygota | Arthropoda |
|---|---|---|---|
| Complete BUSCOs (C) | 1,611 (97.2%) | 2,320 (95%) | 1,044 (98%) |
| Complete and single-copy BUSCOs (S) | 1,129 (68.1%) | 1,573 (64.4%) | 746 (70%) |
| Complete and duplicated BUSCOs (D) | 482 (29.1%) | 747 (30.6%) | 298 (28%) |
| Fragmented BUSCOs (F) | 25 (1.5%) | 67 (2.7%) | 13 (1.2%) |
| Missing BUSCOs (M) | 22 (1.3%) | 55 (2.3%) | 9 (0.8%) |
| Total BUSCO groups searched | 1,658 (100%) | 2,442 (100%) | 1,066 (100%) |

3E, 3G, 3H, 3I, 3P). Indeed, _HeleOBP8_ and _HeleOBP9_ expression levels were significantly higher in males than in females (Figs. 3H and 3I). The _HeleOBP5_ expression level in the male antennae was similar to that in the female antennae but it was significantly higher than all other tissues (Fig. 3E). _HeleOBP21_ expression was similar in the antennae of both sexes, but the expression in female antennae was significantly higher in comparison to all other tissues (Fig. 3P). In addition, _HeleOBP4_, _HeleOBP6_, _HeleOBP11_, _HeleOBP13_, _HeleOBP14_, _HeleOBP15_, _HeleOBP17_ and _HeleOBP18_ showed palp-biased expression (Figs. 3D, 3F, 3J, 3K, 3L, 3M, 3N, 3O), with higher expression of _HeleOBP14_ and _HeleOBP15_ in males than in females. Interestingly, _HeleOBP1_ showed sex-biased expression, exhibiting higher expression in male palps and antennae (Fig. 3B). Finally, _HeleOBP7_ showed higher

## Species Distribution

**Figure 1** Species distribution of unigene sequences of *H. elegans* transcripts relative to other species using homologous BLASTx hits and the NR-NCBI database.

expression in female palps and antennae (Fig. 3G). On the other hand, the expression of CSPs in *H. elegans* was higher in the palps than in the antennae and tibias (Figs. 3Q–3T). Two of the CSPs showed clear sex-biased expression: expression of *HeleCSP1* was higher in male palps (Fig. 3Q), while expression levels of *HeleCSP3* were significantly higher in female palps than in the other tissues (Fig. 3S). The remaining CSPs *HeleCSP2* and *HeleCSP4* showed no differences in the expression values for male and female palps (Figs. 3R and 3T).

### Phylogenetic analyses of OBPs and CSPs

Clades for different types of OBPs were formed, but no one clade formed exclusively with sequences from a single species (Fig. 4). Among the 18 OBP sequences, twelve were classified as Classic, five as Plus-C and one as Dimer. No Minus-C OBPs were identified in the transcriptome. In contrast, the phylogeny for the CSPs (Fig. 5) showed a single clade formed by all four *H. elegans* CSPs and one sequence from *H. oblita*.

## DISCUSSION

Our study provides the first comprehensive reference transcriptome for *Hylamorpha elegans*, obtained from antennae, maxillary palps and right hindleg tibia from males and

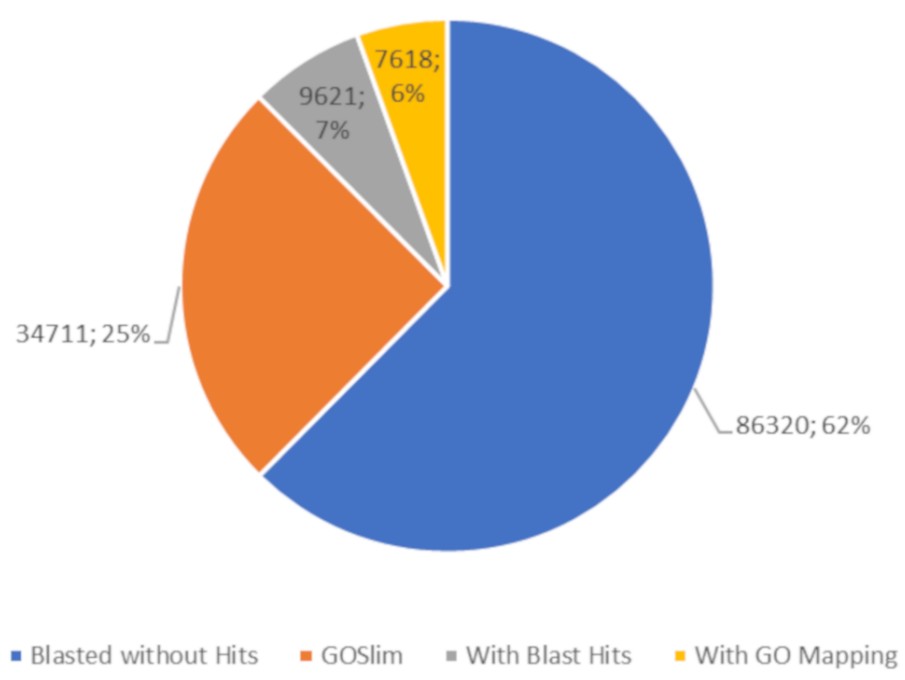

## Annotation Distribution

- Blasted without Hits
- GOSlim
- With Blast Hits
- With GO Mapping

**Figure 2** **Pie chart for the distribution of the Blast2GO annotations.**

females. The number of unigenes obtained from the current transcriptome were similar to that reported for *Holotrichia parallela* and *Anomala corpulenta*, other agriculturally important scarab beetle pests (*Ju et al., 2014*; *Li et al., 2015*; *Yi et al., 2018*). These numbers also resembled those obtained from the transcriptomes of other beetle species, such as *Tomicus yunnanensis* (*Zhu, Zhao & Yang, 2012*), *Ips typographus*, *Dendroctonus ponderosae* (*Andersson et al., 2013*), *Tenebrio molitor* (*Liu et al., 2015*), *Brontispa longissima* (*Bin et al., 2017*), *Callosobruchus chinensis* (*Zhang et al., 2017*) and *Anoplophora chinensis* (*Sun et al., 2018*). Furthermore, BUSCO completeness analysis showed that 98.7% conserved genes were found in this assembly, while the fraction of missing BUSCOs are quite low (1.3%). Although a large percentage of genes indeed did not have GO term association assignment (69.39%), the number of genes with GO term association assignment (30.61%) is higher compared to percentages of GO annotations reported for antennal transcriptomes in other non-model species (10.32% for *Phenacoccus solenopsis*, 9.77% for *Aenasius bambawalei*, (*Nie et al., 2018*); 13.54% for *Cylas formicarius* (*Bin et al., 2017*). Thus, we conclude that this *de novo* transcriptome assembly is suitable for both transcript/gene discovery and gene expression analysis (*Briones et al., 2018*).

From the comparisons of the three tissues studied in both sexes, we identified a higher abundance of all predicted OBP and CSP transcripts in the antennae and/or palps than in the tibia, suggesting that all of these transcripts may take part in chemosensory

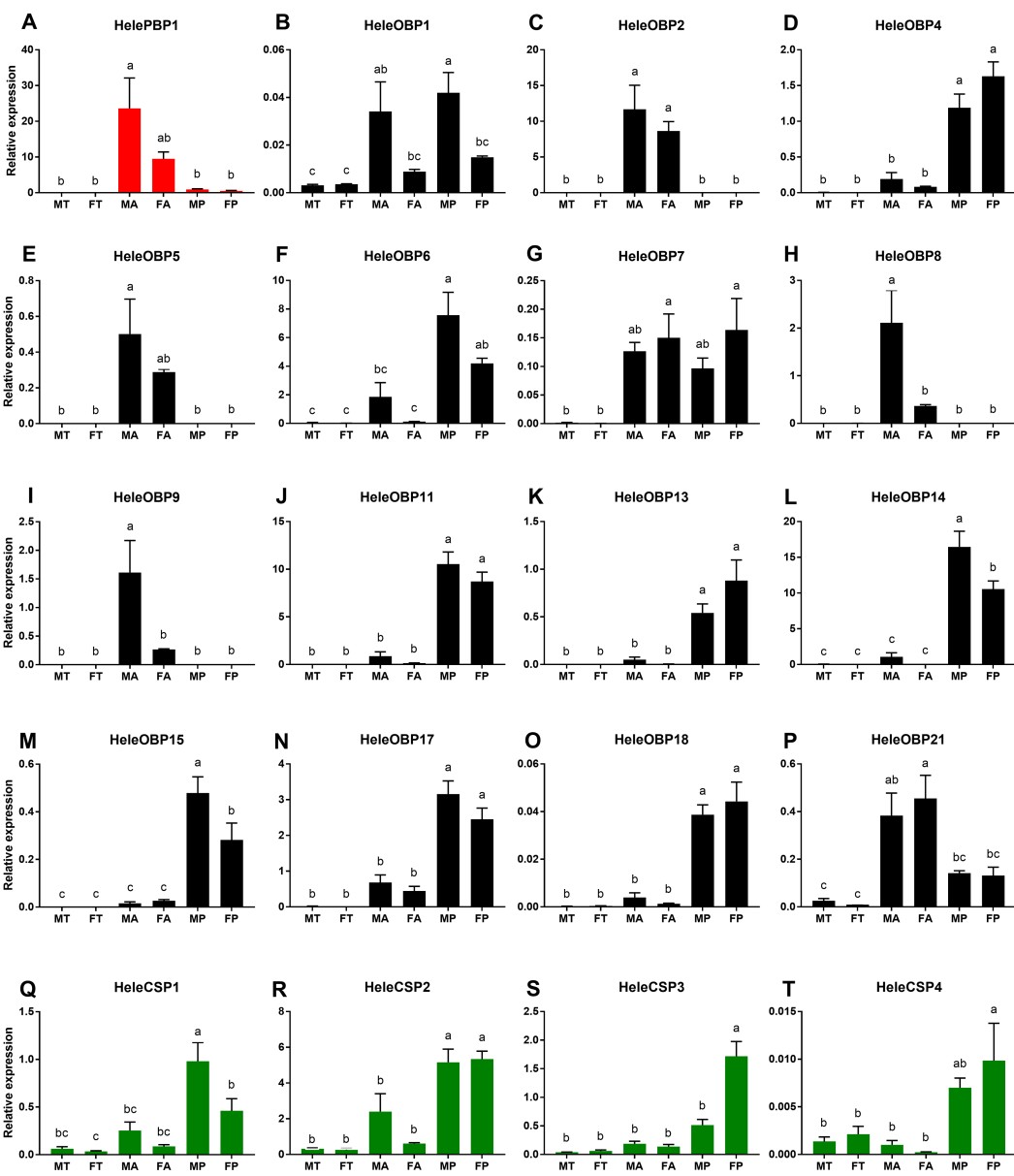

**Figure 3** **Relative expression levels of the putative chemosensory transcripts of *H. elegans* using qRT-PCR.** (A) *H. elegans* PBP1; (B–P) *H. elegans* OBPs; (Q–T) *H. elegans* CSPs. The *x*-axis shows the different tissues. The *y*-axis shows the relative expression level. TM, male tibias; TF, female tibias; AM, male antennae; AF, female antennae; TM, male palps; PF, female palps. Different letters indicate differences according to Tukey-HSD test ($p \leq 0.05$).

processes. Interestingly, the expression levels of several transcripts showed significant differences between males and females. These sex-biased expression patterns suggest that chemosensory genes with higher expression levels in females would play a significant role in host-searching and oviposition behavior (*Nie et al., 2018*), while chemosensory genes displaying higher expression in males would participate in courtship behavior or female

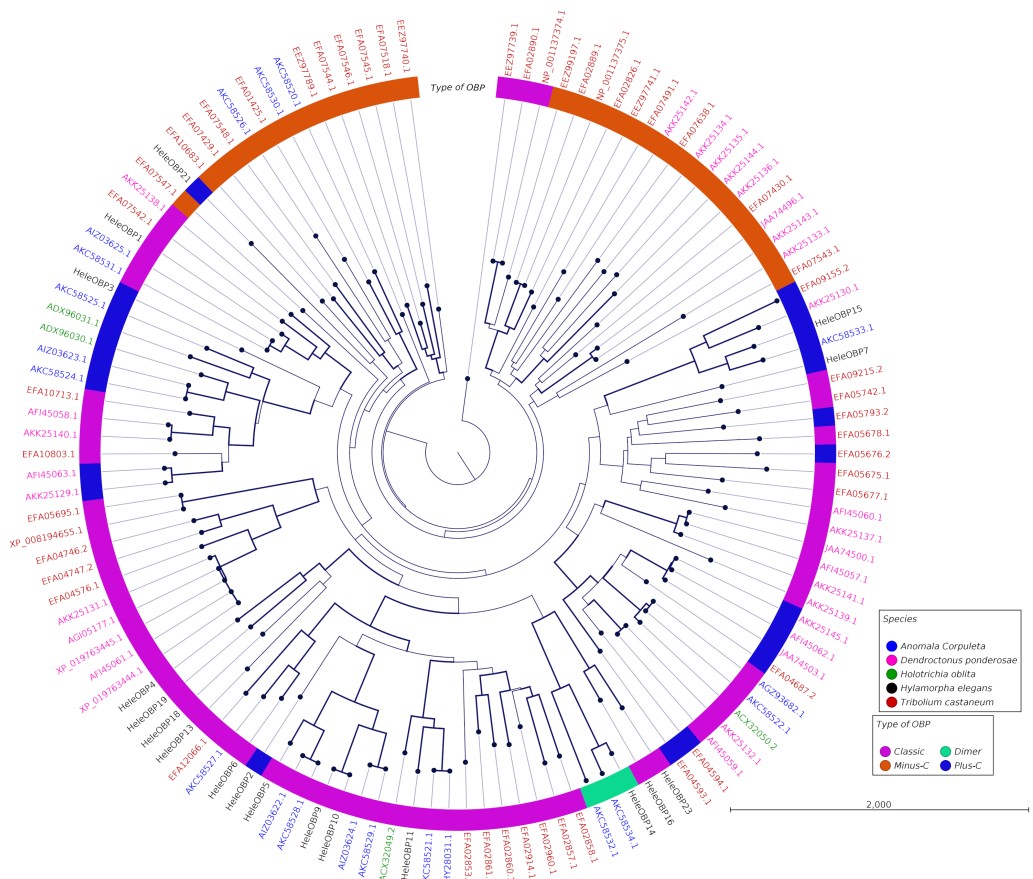

**Figure 4** **Phylogenetic relationship of candidate OBPs from *H. elegans* and other coleopteran species.** The tree was built using the maximum-likelihood method with CLC Workbench v7.8.1 with 1,000 replications. Different color arches indicate different types of OBP classification. Different letter colors indicate the species. OBP names for species other than *H. elegans* are assigned by NCBI ID numbers. Reference line indicates tree scale.

recognition process (*Zhou et al., 2014a*; *Zhou et al., 2014b*). Besides *HelePBP1*, *HeleOBP8* and *HeleOBP9* showed male-biased expression patterns. (Figs. 3H and 3I). Similarly, three OBPs in *Tenebrio molitor* were male antenna-biased, suggesting their role in female recognition (*Liu et al., 2015*). The ability of OBPs to bind sexual pheromones has been reported in the moth species *Amyelois transitella* and *Spodoptera litura* (*Liu et al., 2010*; *Liu, He & Dong, 2012*). These results suggest that further studies of these would reveal their possible role in the sexual behavior of *H. elegans.* Interestingly, among the OBPs, *HeleOBP4, -6, -11, -13, -14, -15, -17,* and *-18* were differentially and more expressed in the palps than the other tissues (Fig. 3). While the relative expression of OBPs has been linked to olfaction roles in antennae, in *Tenebrio molitor,* only eight out of the 19 OBPs were predominantly expressed in the antennae (*Liu et al., 2015*). Given that insects were collected in the field directly from the foliage of their preferred host (*Nothofagus obliqua*) where the insects congregate and feed, it is expected that those OBPs could participate
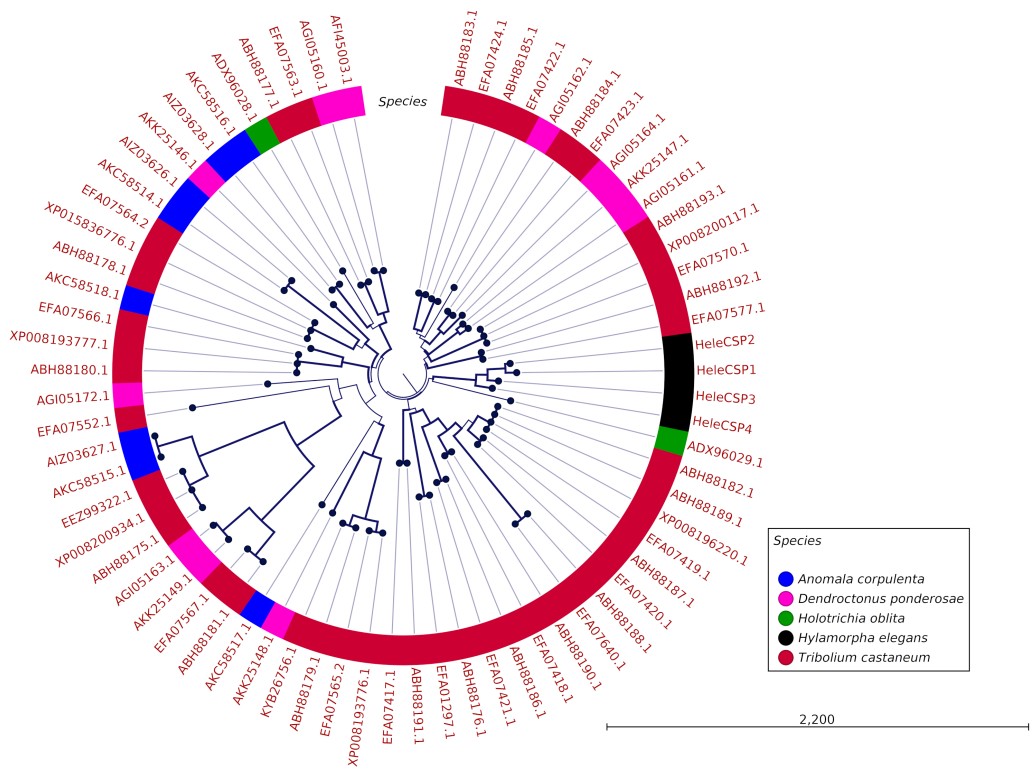

**Figure 5** **Phylogenetic relationship of candidate CSPs from *H. elegans* and other coleopteran species.** The tree was built using the maximum-likelihood method with CLC Workbench v7.8.1 with 1,000 replications. Different color arches indicate different types of OBP classification. Different letter colors indicate the species. OBP names for other species than *H. elegans* are assigned by NCBI ID numbers. Reference line indicates tree scale.

in close-range recognition. Another explanation might be the reduced importance of long-range signals in adults' host-seeking process. Both alternatives should be tested in further experiments.

The total number of CSPs found in different insect species varies widely (*Pelosi et al., 2014*), but the low number of CSPs reported here is in agreement with that reported for the scarab beetle *Anomala corpulenta* (*Li et al., 2015*). Further studies, including other non-sensory tissues such as wings and pheromone glands, may increase the quantity of the reported CSPs (*Sheng et al., 2017*) in addition to studying the expression of CSPs in other developmental stages (*Li et al., 2015*). However, it is important to note that all four CSPs reported here exhibited palp-biased expression. Moreover, the expression of *HeleCSP1* was significantly greater in males, suggesting a possible role in sexual recognition. Previous studies of CSPs have shown that they can bind pheromone components in moth species such as *Mamestra brassicae* (*Jacquin-Joly, 2001*) and that CSPs are able to bind sex pheromones as well as host-plant volatiles in *Sesamia inferens* (*Zhang et al., 2014*).
The expression of *HeleCSP3* was significantly greater in female palps (Fig. 3S). *Ozaki et al. (2008)* assume that female-biased CSP expression is involved in chemoreception and that CSPs transport ligands important to the oviposition behavior of the butterfly *Papilio xuthus*. Furthermore, CSP-silencing produced significant oviposition reduction in the moth *Spodoptera exigua* (*Gong et al., 2012*). Considering these findings, *HeleCSP1* and *HeleCSP3* could be interesting subjects for further studies to establish their roles in male and female behavior.

Phylogenetic analysis for OBPs resulted in clades mostly formed by orthologous sequences grouped by the type of OBP. In *H. elegans,* most of the OBPs were classified as Classic OBPs, which have been found in all insect species studied (*Sánchez-Gracia, Vieira & Rozas, 2009*). Unlike *T. castaneum*, (*Abdel-Latief, 2007*) *A. corpulenta* (*Chen et al., 2014*) and *D. ponderosae* (*Andersson et al., 2013*) *H. elegans* did not show Minus-C OBPs.

Finally, the five Plus-C sequences were highly divergent (Fig. S2), circumstances that could have arisen by a gene duplication mechanism and rapid evolution, resulting in low sequence identity among the members of this group (*Zhou et al., 2004*). For CSPs, the phylogenetic analysis produced clades containing paralog genes (Fig. 5). This pattern agrees with the birth-and-death evolutionary model reported for OBP and CSP families (*Eirín-López et al., 2012*), where paralogous genes have higher divergence times than those of orthologous genes (*Vieira & Rozas, 2011*).

## CONCLUSIONS

The current work reports the first transcriptome for a native scarab beetle from Chile and provides information about the annotation of 165 putative proteins related to gustatory and olfactory processes, helping to increase the number of reported sequences for this important insect family. Our initial hypotheses were confirmed: four out of four studied CSP's transcripts were more abundant in palps, and six out of 16 studied OBP's transcripts were remarkably more abundant in the antennae and eight OBPs were more abundant palps respect to the tissue used as a reference. Additionally, two out of four CSPs and three OBPs showed sex-biased expressions; thus, these transcripts become interesting subjects for further research in *Hylamorpha elegans.*

### Funding

This work was funded by Iniciativa Científica Milenio (ICM) NC120027, Fondecyt Grant 1131008 and Fondecyt for Iniciation in Research Grant 11150721. The funders had no role in study design, data collection and analysis, decision to publish, or preparation of the manuscript.

### Grant Disclosures

The following grant information was disclosed by the authors:
Iniciativa Científica Milenio (ICM): NC120027.

Fondecyt: 1131008.
Fondecyt for Iniciation in Research: 11150721.

## Competing Interests

The authors declare there are no competing interests.

## Author Contributions

- Angélica González-González conceived and designed the experiments, performed the experiments, analyzed the data, prepared figures and/or tables, authored or reviewed drafts of the paper, approved the final draft.
- María E. Rubio-Meléndez performed the experiments, analyzed the data, prepared figures and/or tables, authored or reviewed drafts of the paper, approved the final draft.
- Gabriel I. Ballesteros performed the experiments, analyzed the data, prepared figures and/or tables, authored or reviewed drafts of the paper, approved the final draft.
- Claudio C. Ramírez conceived and designed the experiments, contributed reagents/materials/analysis tools, authored or reviewed drafts of the paper, approved the final draft.
- Rubén Palma-Millanao conceived and designed the experiments, performed the experiments, analyzed the data, contributed reagents/materials/analysis tools, prepared figures and/or tables, authored or reviewed drafts of the paper, approved the final draft.

## DNA Deposition

The following information was supplied regarding the deposition of DNA sequences:

Raw transcriptome data are available in NCBI's Sequence Read Archive database under ID: SRP137879, and Figshare: Palma-Millanao, Ruben; González-González, Angélica; Ballesteros, Gabriel; Ramirez Rivera, Claudio; Rubio Meléndez, Maria Eugenia (2018): Hylamorpha elegans transcriptome assembly. figshare. Dataset. https://doi.org/10.6084/m9.figshare.7271624.v1.

## Data Availability

Data is available at Figshare database
ID: 7271624.v1
URL: Available at https://doi.org/10.6084/m9.figshare.7271624.v1.

## Supplemental Information

Supplemental information for this article can be found online at http://dx.doi.org/10.7717/peerj.7054#supplemental-information.

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
