# Peer review of "Sex- and tissue-specific expression of odorant-binding proteins and chemosensory proteins in adults of the scarab beetle Hylamorpha elegans (Burmeister) (Coleoptera: Scarabaeidae)"

_PeerJ, doi:10.7717/peerj.7054_

## Round 0.1 · original submission · Major Revisions

Dear Dr. González-González and colleagues:

Thanks for submitting your manuscript to PeerJ. I have now received two independent reviews of your work, and as you will see, the reviewers raised some concerns about the research. Despite this, these reviewers are optimistic about your work and the potential impact it will have on research communities studying odorant-binding proteins and chemosensory proteins, as well as scarab beetle biology. Thus, I encourage you to revise your manuscript accordingly, taking into account all of the concerns raised by both reviewers.

Please provide the number of replicates used in RNA-seq experiments. Please also ensure that all of your methods contain enough information such that they are repeatable. Please also ensure that your tree-building is estimation of phylogeny (or just distance-based clustering).

While the concerns of the reviewers are relatively minor, this is a major revision to ensure that the original reviewers have a chance to evaluate your responses to their concerns.

I look forward to seeing your revision, and thanks again for submitting your work to PeerJ.

Good luck with your revision,

-joe

Reviewer 1 ·

Basic reporting

The manuscript is written well, however one more round of proofreading is required (typos, missing articles, etc.). The Introduction provide sufficient background. The Discussion section is too short and should be expanded. Overall, discussion section is too conclusive and does not provide significant insight (the same information was already covered in Results). Authors might consider merging this section with Results. Next, it is not enough to just state that results were similar to other studies. Authors should elaborate, providing quantitative comparison.

Experimental design

Experimental design seems to be appropriate and fit to the objective of the study. Sufficient details are provided.

Validity of the findings

Conclusions are too general (more details are needed). For example, authors might consider adding the information about the main patterns they have observed and discuss the difference observed for different tissues in different sexes.

Reviewer 2 ·

Basic reporting

Basic reporting is sound, and writing is clear and professional.

In the abstract, and throughout the manuscript, statements based on correlation should use appropriate language. For example, "...suggesting that they are involved in the sexual recognition process..." is not based on any direct evidence, so it sounds more assertive than the evidence provided. Similarly for, "that this CSP participates in the oviposition
process".

Lines 19-21: How does transcriptome help to study cryptic habits? The rational is not very clear, and should be expanded or edited.

Experimental design

How many replicates were done for RNA-seq? The information about biological replicates should be added to materials and methods. Without enough biological replicates, the de-novo assembly will be less robust.

Line 162-164: "NCBI (ncbi.nlm.nih.gov) protein 163 sequences of OBPs, CSPs, GRs, ORs, IRs, ORCo, SNMPs, ODEs and PDEs from 954 insect 164 species were downloaded to build databases." How exactly was that done? More details should be added to the methods.

Validity of the findings

No comments.

Additional comments

Figure 3, 4: A Scale should be added to the tree to make it a phylogeny. Otherwise it would just represent the relationship.

Authors should discuss why a large % of genes have no GO terms association assignment?

---

## Round 0.2 · accepted · Accept

Dear Dr. González-González and colleagues:

Thanks for revising your manuscript based on the minor concerns raised by the reviewers. I now believe that your manuscript is suitable for publication. Congratulations! I look forward to seeing this work in print, and I anticipate it being an important resource for research communities studying odorant-binding proteins and chemosensory proteins, as well as scarab beetle biology. Thanks again for choosing PeerJ to publish such important work.

Best,

-joe

Reviewer 1 ·

Basic reporting

Authors have addressed raised issues.

Experimental design

Authors have addressed raised issues.

Validity of the findings

Authors have addressed raised issues.

Reviewer 2 ·

Basic reporting

Both English and scientific clarity has improved in the revised version. The phylogeny figures have also been improved.

Experimental design

Authors provided more details on how sequences were collected and clarified the library construction method, improving the reproducibility of the results.

Validity of the findings

No comments

Additional comments

It will be better to include a brief explanation on the missing GO terms in the methods, and mention that the replicates are not 'true' biological replicates.